# The Impact of Polycentric Structure on $CO_2$ Emissions: Evidence from China

**Jing Wen [1], Xin Zhang [2], Wenjie Du [1], Xiaoying Ouyang [3,4,*] and Zhongchang Sun [3,4,5]**

[1] School of Urban and Regional Science, East China Normal University, Shanghai 200241, China; elainewen1031@163.com (J.W.); duwj1031@126.com (W.D.)
[2] School of Economics and Management, Zhejiang Sci-Tech University, Hangzhou 310018, China; 2020333503219@mails.zstu.edu.cn
[3] Key Laboratory of Digital Earth Science, Aerospace Information Research Institute (AIR), Chinese Academy of Sciences (CAS), Beijing 100094, China; sunzc@aircas.ac.cn
[4] International Research Center of Big Data for Sustainable Development Goals (CBAS), Beijing 100094, China
[5] Key Laboratory Earth Observation Hainan Province, Sanya 572029, China
[*] Correspondence: ouyxy@aircas.ac.cn

**Abstract:** Driven by the 2030 Agenda for Sustainable Development, the importance of sustainable urbanization has taken center stage. In this study, we investigate the impact of polycentric structures on $CO_2$ emissions using data from 279 Chinese cities and employing two-way fixed effects complemented by instrumental variables. Our findings indicate that polycentric structures effectively alleviate $CO_2$ emissions. We identify two key pathways through which polycentric structures contribute to $CO_2$ reduction: promoting green technology and curbing energy consumption. Additionally, we discover that these relationships are influenced by market integration levels and resource dependency. This research offers valuable insights into the future development of sustainable urban spatial structures, paving the way for more eco-friendly cities around the globe.

**Keywords:** Sustainable Development Goals; $CO_2$ emissions; polycentric structure; urban spatial structure; cities

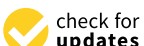



## 1. Introduction

The 2030 Agenda for Sustainable Development offers a visionary roadmap for a sustainable future. There is global consensus on coordinating spatial expansion, economic growth, and the environment to achieve sustainable development [1,2]. Sustainable Development Goal 11 (SDG 11) aim to create inclusive, safe, resilient, and sustainable cities and communities for all. However, the increasing threat of climate change looms over our living environments. The most recent progress report on SDG 13, which focuses on climate action, highlights the urgent need to reduce $CO_2$ emissions, as they reached a historic high in 2021 [3]. In countries like China, rapid urbanization and the concentration of resources in major cities exacerbate $CO_2$ emissions. Consequently, finding a balance between resource integration and sustainable development has become a critical challenge in urban planning.

Research has shown that adopting a polycentric structure can be beneficial in terms of resource integration and optimizing efficiency [4]. Thus, clarifying the impact of polycentric structure on $CO_2$ emissions can help direct the practice of sustainable development. However, the relationship between polycentric structure and $CO_2$ emissions has not reached consensus. Some studies prove that the polycentric structure takes advantage of the integration of resources and optimizes the efficiency of factors [5]. Thus, it can alleviate $CO_2$ emissions [4,6]. It's important to note that some studies have raised concerns about polycentricity, suggesting that it may lead to higher household emissions, increased industrial manufacturing [7], and greater household heating demands [8], all of which could

contribute to higher $CO_2$ emissions. While the other studies demonstrate a non-linear relationship between polycentric structure and $CO_2$ emissions [9,10]. This highlights the need for further exploration and discussion to better understand the intricacies of this relationship.

The inconsistencies in the relationship between polycentricity and $CO_2$ emissions can be explained by four reasons: First, this relationship stems from a balance among diverse influence pathways. Previous studies have examined pathways such as reduced commuting duration, traffic congestion, household carbon emissions, and industrial emissions [7,11]. Focusing on different pathways could lead to inconsistent findings regarding the connection between polycentric structures and $CO_2$ emissions. Second, the sample areas vary from study to study. For example, Burgalassi and Luzzati [7] found that polycentric structure aggravates $CO_2$ emissions in Italy, while Sun et al. [6] and Sha et al. [11] discovered a positive association between polycentricity and $CO_2$ mitigation in China. One plausible reason is that the impact of polycentric structures on $CO_2$ emissions may be influenced by factors such as regional resource dependence and the level of market integration. Third, even within the same country, diverse scales of study regions may lead to a different association between polycentric structures and $CO_2$ emissions. The evidence from Chinese cities showed a negative association between polycentric structure and $CO_2$ [11], while the evidence from Chinese provinces demonstrated a U-shaped relationship [9]. Fourth, the measurement of core variables and the robustness of methods can also influence the results. The influence of the morphological polycentric index differs from that of the functional dimensions of polycentric structures. Additionally, conducting multiple robust tests is recommended to mitigate endogeneity and determine more accurate associations between spatial structures and $CO_2$ emissions [6].

This study evaluates the impact of morphological polycentricity on $CO_2$ emissions in China using a panel dataset from 279 cities spanning from 1999 to 2017. Then we propose two plausible pathways, green technology and energy consumption, and discussed the heterogeneous effects of market integration and resource dependence. Notably, we introduce an innovative instrument variable (IV) in conjunction with a widely used IV to mitigate the endogeneity, exclude the interferences of policies, adopt propensity score matching (PSM) to avoid selection bias, alter fixed effects, clustered levels, and core variable measurements to ensure the robustness of our findings. The contributions of this study are as follows: First, we provide robust evidence of the impact of polycentric structures on $CO_2$ emissions in China, accounting for potential biases from endogeneity and policy-related factors. Second, we discuss the plausible mechanisms that influence this relationship by examining the mediating effects of green technology and energy consumption, thus enhancing our understanding of the spatial structure and $CO_2$ emissions framework. Third, we highlight the heterogeneous effects of market integration and resource dependence, which partly explains the inconsistent results among previous studies.

The rest of the study is organized as follows: Section 2 reveals the data and methods adopted in this study. Section 3 presents the impact of polycentric structure on $CO_2$ emissions and shows the robustness, mechanisms, and heterogeneity of this relationship. Section 4 analyzes the results above. Section 5 summarizes the conclusion and its policy implications. The conceptual framework is represented in Figure 1.

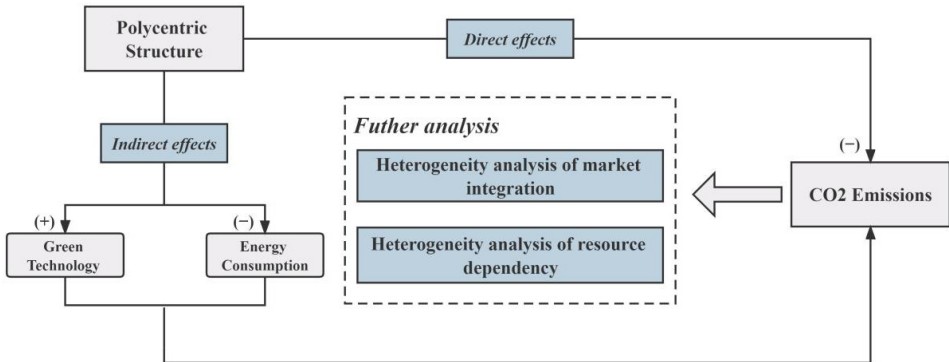

**Figure 1.** Conceptual framework.

## 2. Materials and Methods

### 2.1. Data and Variable

#### 2.1.1. Dependent Variables

We adopted the logarithm of regional $CO_2$ emissions as the dependent variable. The data on $CO_2$ emissions were generated by Chen et al. [12], who used a particle swarm optimization-back propagation algorithm to unify the scale of DMSP/OLS and NPP/VIIRS satellite imagery at the county level. We carried out our study using the data of the $CO_2$ emissions between 1999 and 2017.

#### 2.1.2. Independent Variable

In this study, we focused on morphological polycentricity, which is a balanced distribution of the importance of the cities within a given region. Generally, it can be quantified by employment, population size, or economic output [6]. We used the Prolonged Artificial Nighttime-Light Dataset of China generated by Zhang et al. [13] to measure the economic activity of cities [10]. Based on this dataset, we measured the polycentricity as the Pareto exponent, which is calculated by the slope of the size and the ranks of each county within a city [14]:

$$\ln\left(Rank_j\right) = A - QlnLight_j \tag{1}$$

where $Q$ refers to the level of polycentricity. The higher the $Q$-value, the more polycentric and compact a city's spatial structure is. $Rank_j$ is the rank of economic activity of county $j$ in descending order within the given city. $lnsize_j$ is the economic activity level of county $j$.

#### 2.1.3. Control Variables

Following previous studies on $CO_2$ emissions, we controlled secondary industry as a percentage of GDP, foreign direct investment, population density, the percentage of budget fiscal expenditure and technology expenditure on GDP, and the ratio of the employed population to exclude their influence on $CO_2$ emissions [15,16]. Table 1 defined all the variables used in this study.

### 2.2. Data Summary

Descriptive statistics for each variable were reported in Table 2. The overall $CO_2$ emissions had a value of 2.743 on average. The average polycentricity was 0.924. Generally, if the $q$-value is greater than 1, the spatial structure of the city is polycentric. If it is less than 1, the spatial structure is relatively monocentric.

**Table 1.** Definitions and operations of variables.

| Variable | Definition |
| --- | --- |
| Dependent variable | |
| Carbon | The logarithm of regional $CO_2$ emissions. |
| lndependent variables | |
| Q | The Pareto exponent. |
| Mediating variable | |
| Gti | The logarithm of the annual number of green patent applications plus one. |
| Energy | The logarithm of electricity consumption. |
| Control variables | |
| lngy | The share of the secondary sector in GDP. |
| lnFDI | The logarithm of the ratio of the region's actual use of foreign capital to GDP in that year. |
| lnpeople | The logarithm of the ratio of the total population to the local land area at the end of the year. |
| lnfis | The logarithm of the ratio of regional general budget fiscal expenditure to GDP. |
| lnsci | The logarithm of the ratio of regional science and technology expenditure to GDP. |
| lnurba | The logarithm of the ratio of the urban employed population to the year-end household population. |
| Other variables | |
| Hlhl | The ratio of the river density to the exchange rate. |
| Unemploy | The ratio of unemployment to the population. |
| LCC | If it is the pilot city of Low-carbon city policy, this variable = 1, 0 otherwise. |
| ETS | If it is the pilot city of the emissions trading scheme, this variable = 1, 0 otherwise. |
| Rep_carbon | The $CO_2$ emissions proposed by of Wu et al. [17] |
| PCE | $CO_2$ emissions per capita. |
| Lngreencov | The logarithm of canopy coverage. |
| Lnlpg | The logarithm of household consumption of liquefied petroleum gas. |
| Lnws | The logarithm of average wind speed. |
| Lnhum | The logarithm of average relative humidity. |
| Lntem | The logarithm of average temperature. |

### 2.3. Methodology

Accounting for unit-variant and time-variant variables, we adopted a two-way fixed effects model to control for both unit- and time-fixed effects. We constructed the following model:

$$Carbon_{i,t} = \alpha_0 + \alpha_1 Q_{i,t} + \sum \tau_j * X_{j,i,t} + \mu_i + \varphi_t + \varepsilon_{i,t} \qquad (2)$$

where the subscript $i$ represents different cities, $t$ represents time variables, $Carbon_{i,t}$ represents the natural logarithm of regional $CO_2$ emissions, $Q_{i,t}$ represents the polycentricity of city $i$, $X_{j,i,t}$ is the matrix-vector of the control variables, $\mu_i$ represents a city-specific effect to control for unobserved heterogeneity, $\varphi_t$ represents the control for the year-fixed effect, and $\varepsilon_{i,t}$ is the stochastic disturbance term.

To mitigate concerns about endogeneity and figure out the causality between polycentric spatial structures and $CO_2$ emissions, we constructed two IVs that were correlated with polycentric spatial structures but met the exogeneity condition of $CO_2$. Specifically, we followed Chen et al. [10] and used the river density/exchange rate (i.e., Hlhl) as the first instrument for polycentricity [10]. Proximity to water sources could affect the spatial structure of cities [18]. In the meantime, it is not related to $CO_2$ emissions. Thus, river density is a proper IV for polycentricity.

**Table 2.** Descriptive statistics of variables.

| Variables | Obs. | Mean | S.D. | Min | Max |
|---|---|---|---|---|---|
| Carbon | 5278 | 2.743 | 0.858 | 0.0881 | 5.441 |
| Q | 5269 | 0.924 | 0.540 | 0.252 | 5.498 |
| lngy | 5226 | 3.868 | 0.282 | 2.086 | 4.525 |
| lnFDI | 4672 | −6.201 | 1.386 | −15.250 | −1.816 |
| lnpeople | 5295 | −2.717 | 0.973 | −7.624 | 0.340 |
| lnfis | 5290 | −2.094 | 0.504 | −4.221 | 0.994 |
| lnsci | 5279 | 11.370 | 1.270 | 5.727 | 15.430 |
| lnurba | 5240 | −1.809 | 0.665 | −3.358 | 1.249 |
| Hlhl | 5330 | 0.291 | 0.209 | 0.00577 | 2.278 |
| Unemploy | 5278 | 3.884 | 0.697 | −0.486 | 7.051 |
| LCC | 5330 | 0.139 | 0.346 | 0 | 1 |
| ETS | 5330 | 0.0356 | 0.185 | 0 | 1 |
| Rep_carbon | 5071 | 7.841 | 0.569 | 6.047 | 11.51 |
| PCE | 5275 | −3.104 | 0.765 | −5.931 | −0.270 |
| Win_carbon | 5278 | 2.742 | 0.844 | 0.739 | 4.652 |
| Gti | 4968 | 3.225 | 1.683 | 0.693 | 8.897 |
| Energy | 2789 | 13.070 | 1.221 | 4.625 | 16.54 |
| Lngreencov | 5243 | 3.488 | 0.459 | −1.715 | 5.957 |
| Lnlpg | 5125 | 8.820 | 1.510 | 1.386 | 13.809 |
| Lnws | 5330 | 0.735 | 0.266 | −0.302 | 1.582 |
| Lnhum | 5330 | 4.227 | 0.143 | 3.590 | 4.481 |
| Lntem | 5323 | 2.588 | 0.482 | −2.036 | 3.274 |

We then proposed an innovative instrument, namely, the proportion of unemployment (i.e., unemployment as a percentage of the population). It has been proven that there are interdependencies between labor market behavior and the spatial structure of cities [19]. A large proportion of unemployed people indicates that the city has developed excessively or poorly, which leads to a lack of impetus for further expansion of the city, which has a negative impact on polycentricity. Meanwhile, the proportion of unemployment cannot affect $CO_2$.

Finally, the ratio of river density to exchange rate and the proportion of unemployment were IVs for polycentric spatial structure. The first stage of two-stage least squares (2SLS) was as follows:

$$Q_{i,t} = \beta_0 + \beta_1 Hlhl_{i,t} + \beta_2 Unemploy_{i,t} + \sum \tau_j * X_{j,i,t} + \mu_i + \varphi_t + \varepsilon_{i,t} \qquad (3)$$

where $Hlhl_{i,t}$ and $Unemploy_{i,t}$ represent the ratio of river density to the exchange rate and the proportion of unemployment, respectively.

The second stage of 2SLS was as follows:

$$Carbon_{i,t} = \delta_0 + \delta_1 \widehat{Q_{i,t}} + \sum \tau_j * X_{j,i,t} + \mu_i + \varphi_t + \varepsilon_{i,t} \qquad (4)$$

where $\widehat{Q_{i,t}}$ is the first-stage predicted value for the polycentricity (i.e., $Q_{i,t}$).

Previous studies have verified the mediating role of transportation, household carbon emissions, and industrial emissions in the relationship between polycentric structures and $CO_2$ emissions [6,11]. We proposed two innovative pathways to examine how polycentric structures affect $CO_2$ emissions. We hypothesized that green technology and energy consumption might serve as potential mechanisms through which polycentric structures influence $CO_2$ emissions. To test these hypotheses, we measured green technology as the logarithm of the annual number of green patent applications plus one assessing its mediating effect on the relationship. For energy consumption, we used the logarithm of electricity consumption to evaluate its potential mediating role in the relationship between polycentricity and $CO_2$ emissions. With these variables in place, we constructed the

following model to analyze the impact of polycentric structures on $CO_2$ emissions through these proposed pathways:

$$M_{i,t} = \gamma_0 + \gamma_1 Q_{i,t} + \sum \tau_j * X_{j,i,t} + \mu_i + \varphi_t + \varepsilon_{i,t} \tag{5}$$

where $M_{i,t}$ represents mediators (i.e., green technology and energy consumption).

## 3. Results

### 3.1. Effect of the Polycentric Spatial Structures on $CO_2$ Emissions

Based on Equation (1), Columns (1)–(3) of Table 3 reported the results with a two-way fixed effect with robust standard errors, Driscoll-Kraay standard errors [20], and 1000 bootstrap replications, respectively. The results demonstrated a significant negative effect of polycentricity on $CO_2$ emissions, which implied that polycentric spatial structures helped reduce $CO_2$ emissions in China.

**Table 3.** The impact of polycentric spatial structures on $CO_2$ emissions.

| Variables | (1) FE Carbon | (2) FE_DK Carbon | (3) FE_BS Carbon | (4) First Stage Q | (5) Second Stage Carbon |
|---|---|---|---|---|---|
| Q | −0.050 *** (−2.61) | −0.050 *** (−3.32) | −0.050 ** (−2.48) | | −0.772 *** (−4.06) |
| Hlhl | | | | 5.233 *** (3.57) | |
| Unemploy | | | | −0.029 *** (−4.31) | |
| Control | YES | YES | YES | YES | YES |
| City FE | YES | YES | YES | YES | YES |
| Year FE | YES | YES | YES | YES | YES |
| Observations | 4513 | 4513 | 4513 | 4496 | 4496 |
| R-squared | 0.947 | 0.947 | 0.947 | | 0.882 |
| K-P LM statistic | | | | | 24.219 *** |
| K-P Wald F statistic | | | | | 12.663 |
| Hansen J statistic | | | | | 0.219 |

Notes: t/z-values are in parentheses; **, and *** represent the 5%, and 1% significance levels, respectively; the coefficients of the control variables (i.e., the secondary industry as a percentage of GDP, foreign direct investment, population density, the percentage of budget fiscal expenditure and technology expenditure on GDP, and the ratio of the employed population), year and time dummies are not reported to save space; the unit of river/exchange rate is %.

### 3.2. Robustness Checks

#### 3.2.1. Endogeneity

The results based on the 2SLS estimator were presented in Columns (4) and (5) of Table 3. The estimation results of the first stage demonstrated that the associations between IVs and polycentricity were significant at a 1% level, and the F-statistic is 12.663, surpassing the threshold for a strong instrument. The K-P LM statistic was significant at a 1% level, indicating the IVs were related to the polycentric spatial structure. And the Hansen J-test indicated that there were no over-identification problems and the IVs were strictly exogenous.

The results suggested that polycentric spatial structure supports $CO_2$ emission reduction after considering endogeneity. Compared with the coefficients in Columns (1)–(3), the coefficients estimated by 2SLS were greater than those estimated by the two-way fixed effect. One plausible reason was that there might be reverse causality between polycentric spatial structure and $CO_2$ emission because air pollution could also affect the labor market, which would influence the spatial structure of cities [21]. The 2SLS approach helped mitigate this potential source of bias.

### 3.2.2. Alter Fixed Effects and Clustered Levels

In the baseline results, we controlled both individual-fixed effects and year-fixed effects. To avoid the influence of unmeasured factors, we controlled for fixed effects in the dimension of the province. Furthermore, we adopted interactive clustering that took different clustering levels, such as city, province, and the interaction term of these two levels, into account. The results for altering fixed effects and clustering levels were shown in Figure 2. The blue line showed the 95% confidence interval, the orange point presented the coefficient, and the red dotted line indicated the 0 value. The results indicated that polycentric spatial structure had a significantly negative impact on $CO_2$ emissions. The fixed effect of province dimension and diverse cluster levels did not influence this relationship; namely, the benchmark results were robust after considering these unmeasured factors.

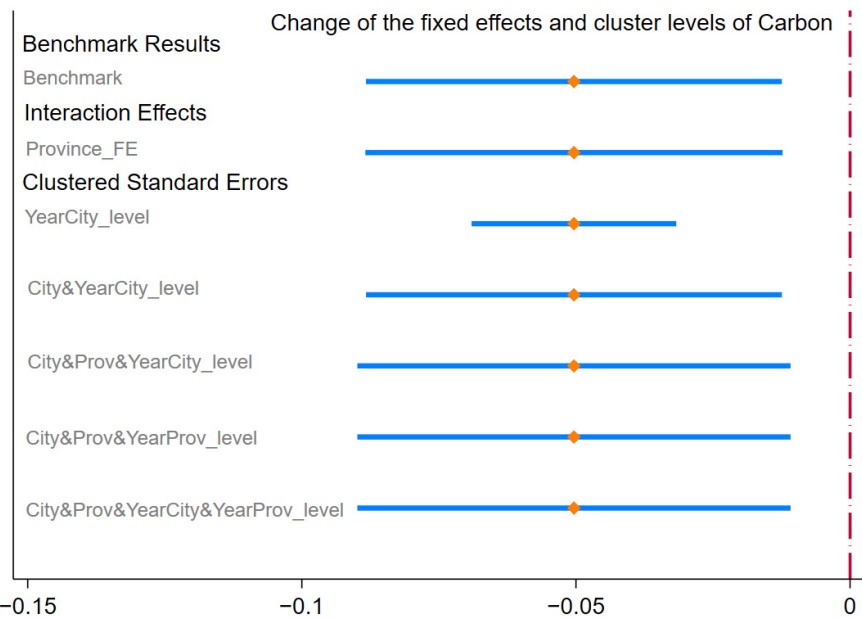

**Figure 2.** Changes in the fixed effects and cluster levels.

### 3.2.3. Exclude the Interference of Related Policies

$CO_2$ emissions highly interfered with relevant policies in China. For example, the low-carbon city (LCC) policy emphasized $CO_2$ emissions along with economic development [22]. The effectiveness of LCC in $CO_2$ emissions abatement had been verified in pilot cities [23]. Furthermore, the emissions trading scheme (ETS), as a typical market-oriented environmental instrument, was also conducive to reducing $CO_2$ emissions [24]. In this regard, the interference of LCC and ETS might confound the relationship between polycentricity and $CO_2$ emissions.

Thus, we controlled these two policies based on the benchmark results to mitigate their interference. The results were presented in Columns (1) and (2) of Table 4. After controlling the relevant policies (i.e., LCC and ETS), respectively, the coefficients of polycentricity are still significantly negative. It indicated that polycentric spatial structures helped reduce $CO_2$ emissions; namely, the benchmark results are robust.

### 3.2.4. Alter the Measurement of $CO_2$ Emissions

The impact of polycentricity on $CO_2$ emissions might also be influenced by the measurement of $CO_2$ emissions. To guarantee the robustness of benchmark results, we measured the $CO_2$ emissions by Wu et al. [17] and $CO_2$ emissions per capita. The estimated results were shown in Columns (3) and (4) of Table 4, respectively. The impact of polycentric spatial structure helped $CO_2$ abatement, indicating that altering the measurement of $CO_2$ emissions did not influence the benchmark results.

**Table 4.** Robustness check: the impact of polycentric spatial structures on $CO_2$ emissions.

| Variables | (1) Carbon | (2) Carbon | (3) Rep_Carbon | (4) PCE | (5) Win_Carbon |
|---|---|---|---|---|---|
| Q | −0.045 ** | −0.045 ** | −0.018 ** | −0.128 *** | −0.055 *** |
| | (−2.33) | (−2.42) | (−2.52) | (−2.67) | (−2.92) |
| LCC | −0.067 *** | | | | |
| | (−4.26) | | | | |
| ETS | | −0.119 *** | | | |
| | | (−5.83) | | | |
| Control | YES | YES | YES | YES | YES |
| City FE | YES | YES | YES | YES | YES |
| Year FE | YES | YES | YES | YES | YES |
| Observations | 4513 | 4513 | 4313 | 4513 | 4513 |
| R-squared | 0.948 | 0.949 | 0.850 | 0.915 | 0.941 |

Notes: t-values are in parentheses; **, and *** represent the 5%, and 1% significance levels, respectively; the coefficients of the control variables (i.e., the secondary industry as a percentage of GDP, foreign direct investment, population density, the percentage of budget fiscal expenditure and technology expenditure on GDP, and the ratio of the employed population), year and time dummies are not reported to save space.

Furthermore, the results of a regression could be sensitive to outliers, while the $CO_2$ emissions varied widely among 4513 city-year observations (minimum = 0.0881, maximum = 5.441). Thus, we tailored the data at the 1% level. The results were presented in Column (5) of Table 4. After excluding the outliers in $CO_2$ emissions, the polycentricity still had a significant and negative impact on $CO_2$ emissions, indicating that the benchmark results were robust.

### 3.2.5. PSM

One might argue that the $CO_2$ abatement was not achieved by polycentric spatial structure but by the selection bias between polycentric structure and monocentric structure. To mitigate such selection bias, we adopted propensity scores to match cities in the polycentric structure with those in the monocentric structure and make them comparable. The propensity score is estimated using a logit regression model. Columns (1)–(3) in Table 5 reported the results using nearest neighbor matching (1:4), caliper matching, and kernel matching, respectively. The results of balance tests were reported in Appendix A Table A1 and Figure A1, Table A2 and Figure A2, and Table A3 and Figure A3, respectively. The results of balance tests demonstrated that after matching, there was no significant bias between cities with polycentric and monocentric structures, indicating that PSM helped mitigate selection bias. After matching, the impact of polycentric spatial structures on $CO_2$ emissions was still significantly negative, implying a positive role for polycentricity in $CO_2$ reduction.

**Table 5.** Robustness check: the impact of polycentric spatial structures on $CO_2$ emissions using PSM.

| Variables | (1) Carbon | (2) Carbon | (3) Carbon |
|---|---|---|---|
| Q | −0.0558 *** | −0.0508 *** | −0.0508 *** |
| | (−3.15) | (−2.64) | (−2.64) |
| Control | YES | YES | YES |
| City FE | YES | YES | YES |
| Year FE | YES | YES | YES |
| Observations | 3771 | 4508 | 4508 |
| R-squared | 0.947 | 0.947 | 0.947 |

Notes: t-values are in parentheses; *** represent the 1% significance levels, respectively; population density, the percentage of budget fiscal expenditure and technology expenditure on GDP, and the ratio of the employed population are adopted to match cities in the polycentric structure with those in the monocentric structure; the coefficients of the control variables (i.e., the secondary industry as a percentage of GDP, foreign direct investment, population density, the percentage of budget fiscal expenditure and technology expenditure on GDP, and the ratio of the employed population), year and time dummies are not reported to save space.

### 3.2.6. Exclude Other Confounders

Despite accounting for various confounding factors that could influence the relationship between polycentric structures and $CO_2$ emissions in our baseline results, there were still some potential confounders that we have not considered. Factors such as canopy coverage [25], energy consumption [26], wind speed [27], humidity [28], and temperature [29] may affect carbon emissions. To minimize the impact of these confounders, we included them as controls in our baseline results. Columns (1)–(5) in Table 6 displayed the effect of polycentric spatial structures on $CO_2$ emissions after controlling for canopy coverage, energy consumption, wind speed, humidity, and temperature, respectively. The results indicated that polycentric structures reduce carbon emissions even after accounting for the confounding influences of these additional factors. Column (6) presented the results after controlling for all potential confounders, and the findings remained robust.

**Table 6.** Robustness check: The impact of polycentric spatial structures on $CO_2$ emissions.

| Variables | (1) Carbon | (2) Carbon | (3) Carbon | (4) Carbon | (5) Carbon | (6) Carbon |
|---|---|---|---|---|---|---|
| Q | −0.052 *** | −0.051 ** | −0.050 *** | −0.050 *** | −0.050 *** | −0.051 *** |
|  | (−2.77) | (−2.55) | (−2.60) | (−2.60) | (−2.59) | (−2.67) |
| lngreencov | 0.030 |  |  |  |  | 0.031 |
|  | (1.43) |  |  |  |  | (1.38) |
| lnlpg |  | 0.003 |  |  |  | 0.002 |
|  |  | (0.004) |  |  |  | (0.41) |
| lnws |  |  | −0.005 |  |  | −0.014 |
|  |  |  | (0.047) |  |  | (−0.28) |
| lnhum |  |  |  | −0.162 * |  | −0.162 * |
|  |  |  |  | (−1.91) |  | (−1.81) |
| lntem |  |  |  |  | 0.022 | −0.007 |
|  |  |  |  |  | (0.85) | (−0.30) |
| Control | YES | YES | YES | YES | YES | YES |
| City FE | YES | YES | YES | YES | YES | YES |
| Year FE | YES | YES | YES | YES | YES | YES |
| Observations | 4473 | 4400 | 4513 | 4513 | 4510 | 4364 |
| R-squared | 0.948 | 0.947 | 0.947 | 0.947 | 0.947 | 0.948 |

Notes: t-values are in parentheses; *, **, and *** represent the 10%, 5%, and 1% significance levels, respectively; the coefficients of the control variables (i.e., the secondary industry as a percentage of GDP, foreign direct investment, population density, the percentage of budget fiscal expenditure and technology expenditure on GDP, and the ratio of the employed population), year and time dummies are not reported to save space.

### 3.3. Possible Mechanisms

The analysis above demonstrated that polycentricity had a notable impact on $CO_2$ emissions. We also highlighted the mediating role that green technology and energy consumption played in the relationship between polycentricity and $CO_2$ emissions. By understanding this connection, we could develop a more comprehensive understanding of the factors influencing $CO_2$ emissions and make more informed decisions on sustainable urban development.

The results demonstrated that polycentric structure had a positive impact on green technology (see Columns (1)–(3) in Table 7), while had a negative impact on energy consumption (see Columns (4)–(6) in Table 7). It indicated that polycentric structures promoted green technology and reduced energy consumption, both of which were conducive to $CO_2$ abatement. In this regard, a polycentric structure could reduce $CO_2$ emissions by promoting green technology and reducing energy consumption.

### 3.4. Heterogeneity Analysis

The spatial structures and $CO_2$ emissions were heterogeneous because of the different features in each city. We emphasized the heterogeneity resulting in the administrative hierarchy and the dependence on resources in this context. First, we grouped the sample into municipalities directly under the central government and other cities. Second, we hypothesized that the impact of polycentricity on $CO_2$ varied between resource-based cities and others because of the resource curse. The empirical results were presented in Table 8.

**Table 7.** The impacts of green technology and energy consumption on $CO_2$ emissions.

| Variables | (1) Green Technology FE | (2) FE_DK | (3) FE_BS | (4) Energy Consumption FE | (5) FE_DK | (6) FE_BS |
|---|---|---|---|---|---|---|
| Q | 0.173 *** | 0.173 *** | 0.173 *** | −0.121 ** | −0.121 | −0.121 ** |
|  | (3.10) | (3.55) | (2.85) | (−2.36) | (−1.63) | (−2.21) |
| Control | YES | YES | YES | YES | YES | YES |
| City FE | YES | YES | YES | YES | YES | YES |
| Year FE | YES | YES | YES | YES | YES | YES |
| Observations | 4286 | 4286 | 4286 | 2375 | 2375 | 2375 |
| R-squared | 0.877 | 0.877 | 0.877 | 0.654 | 0.654 | 0.654 |

Notes: t-values are in parentheses; **, and *** represent the 5%, and 1% significance levels, respectively; the coefficients of the control variables (i.e., the secondary industry as a percentage of GDP, foreign direct investment, population density, the percentage of budget fiscal expenditure and technology expenditure on GDP, and the ratio of the employed population), year and time dummies are not reported to save space.

**Table 8.** Heterogeneity analysis for different features of cities.

| Variables | (1) PCM | (2) Non-PCM | (3) Resource-Based | (4) Non-Resource-Based |
|---|---|---|---|---|
| Q | −0.064 | −0.052 *** | −0.020 | −0.055 *** |
|  | (−0.17) | (−2.66) | (−0.55) | (−2.78) |
| Control | YES | YES | YES | YES |
| City FE | YES | YES | YES | YES |
| Year FE | YES | YES | YES | YES |
| Observations | 507 | 4006 | 1693 | 2820 |
| R-squared | 0.920 | 0.951 | 0.945 | 0.949 |

Notes: t-values are in parentheses; *** represent the 1% significance levels, respectively; the coefficients of the control variables (i.e., the secondary industry as a percentage of GDP, foreign direct investment, population density, the percentage of budget fiscal expenditure and technology expenditure on GDP, and the ratio of the employed population), year and time dummies are not reported to save space.

Columns (1) and (2) in Table 8 compared the impact of polycentric structure on the $CO_2$ emissions of municipalities directly under the central government and the others. The results demonstrated that polycentricity had no significant impact on the $CO_2$ emissions of municipalities while reduced the $CO_2$ emissions of other cities. Columns (3) and (4) presented the relationship between polycentric structure and $CO_2$ emissions in resource-based cities and other cities, respectively. The results indicated that the polycentric structure affected $CO_2$ emissions neither in municipalities nor resource-based cities while significantly reduced $CO_2$ emissions in other cities.

## 4. Discussion

Our primary empirical findings suggested that polycentric structures contributed to $CO_2$ abatement, aligning with previous studies that found increased $CO_2$ efficiency [11] and decreased $CO_2$ concentrations [4]. Monocentric structures, which partly represent resource agglomeration in cities, tend to exacerbate emissions [30]. In contrast, polycentric structures can mitigate agglomeration diseconomies, promote market integration, and optimize energy efficiency [10,31], ultimately leading to $CO_2$ reduction. Our results build upon existing knowledge by using panel data from Chinese city regions and conducting comprehensive robustness tests.

Green technology and energy consumption had been identified as potential mechanisms linking polycentric structures to $CO_2$ emissions. On the one hand, polycentric spatial structure is conducive to the green economy, which is inevitable for green technology [32,33]. In the meantime, green technology is widely accepted as being environmentally friendly and helping with $CO_2$ reduction [34]. On the other hand, with the optimization of the polycentric structure, mega-cities form internal subcenters represented by industrial agglomeration, and hence strengthen the level of specialization. Therefore, in the context of coordinated industrial development, the polycentric structure promotes green technology. Furthermore, the polycentric spatial structure might reduce $CO_2$ through energy

consumption. One plausible reason is that polycentric development could affect energy efficiency [35] and, hence, influence energy consumption. Meanwhile, energy consumption is prone to $CO_2$ emissions [36].

Additionally, the impact of polycentric structures on $CO_2$ emissions was influenced by administrative hierarchy and resource dependence. Firstly, polycentric structures had no significant influence on $CO_2$ emissions in municipalities, potentially due to their relatively higher degree of regional market integration. As market segmentation can improve energy efficiency and decrease $CO_2$ emissions [31], cities with strong market integration might not benefit as much from polycentric structures [10]. Secondly, the polycentric structures did not affect $CO_2$ emissions in resource-based cities but reduced emissions in other cities. This can be explained by the detrimental impact of the resource curse on environmental sustainability, as resource-based cities often have a single industry structure, which aggravates $CO_2$ emissions [37]. Resource misallocation decreases energy efficiency and, hence, accelerates $CO_2$ emissions [38]. Polycentric structures upgrade industry structures [39] and hence have a relatively larger impact on cities with diverse industry structures. Namely, it reduces $CO_2$ emissions in cities that are not heavily reliant on resources.

## 5. Conclusions

Fueled by the urgency of achieving the Sustainable Development Goals (SDGs), addressing the rise in $CO_2$ emissions has become a pressing concern. This study delved into the role of sustainable urban development in mitigating $CO_2$ emissions by investigating the impact of polycentric structures on $CO_2$ emissions in 278 Chinese cities from 1999 to 2017. We employed a two-way fixed effect model to estimate our benchmark results and constructed IVs to address endogeneity concerns. Our findings revealed that polycentric structures contributed to $CO_2$ emissions reduction by promoting green technology and lowering energy consumption. Notably, the positive impact of polycentric structures on $CO_2$ emissions was only significant in cities that were not classified as municipalities or resource-based cities.

Our study marginally contributed to the theory. First, we figured out a robust relationship between polycentric structures and $CO_2$ emissions, accounting for potential biases from policy interferences, selection bias, and core variable measurements. Second, we proposed two innovative pathways that mediated the impact of spatial structures on $CO_2$ emissions. Third, we considered the heterogeneous effects of administrative hierarchy and resource dependence in this context.

From our findings, we derived practical implications for achieving SDGs. First, polycentric structures offer a viable approach for developing countries to regulate $CO_2$ emissions. When it comes to China specifically, some cities in China should gradually build polycentric spatial structures. Second, government departments should implement strategies related to green technology and energy consumption to regulate $CO_2$ emissions and achieve SDGs. Third, urban planning and $CO_2$ mitigation strategies should be tailored to individual cities, with a focus on establishing more environmental regulations in resource-based cities and municipalities.

In closing, we acknowledge some limitations and avenues for future research. First, our study used county-level data to calculate the morphological polycentric index at the city level, overlooking the functional dimensions of polycentric structures. Future studies should explore these functional dimensions further. Second, our analysis focused on the Pareto index in relation to city size balance, suggesting that future research could examine the impact of spatial balance on $CO_2$ emissions. Lastly, while our study area is China, future studies could investigate the heterogeneous effects of diverse countries and delve into other aspects of the SDGs to gain a more comprehensive understanding of the topic.

**Author Contributions:** Conceptualization, J.W. and X.Z.; methodology, X.Z.; software, J.W. and X.Z.; formal analysis, W.D.; data curation, X.Z.; writing—original draft preparation, J.W. and W.D.; writing—review and editing, Z.S.; visualization, W.D.; supervision, Z.S. and X.O.; project administration, Z.S. and X.O.; funding acquisition, Z.S. All authors have read and agreed to the published version of the manuscript.

**Funding:** This work was jointly supported by the Innovative Research Program of the International Research Center of Big Data for Sustainable Development Goals (Grant No. CBAS2022IRP04), the National Natural Science Foundation of China (Grant No. 42171291), the Key R&D Program Projects in Hainan Province (grant number ZDYF2020192), and the Key Research and Development Program of Guangxi (Guike-AB22035060).

**Data Availability Statement:** The data on nighttime light are openly available in the Prolonged Artificial Nighttime-Light Dataset of China at https://doi.org/10.11888/Socioeco.tpdc.271202. The data on $CO_2$ emissions are openly available at https://www.ceads.net/data/county/ (accessed on 8 May 2023). Other city-level data are obtained from the China City Statistical Yearbook (1999–2017).

**Acknowledgments:** The authors appreciate the useful discussion of editors and reviewers.

**Conflicts of Interest:** The authors declare no conflict of interest.

## Appendix A

**Table A1.** Balance Test of the Nearest Neighbor Matching (1:4).

| Variables | Unmatched Matched | Mean Treated | Control | Bias (%) | Reduce Bias (%) | t-Test T | P > T |
|---|---|---|---|---|---|---|---|
| lnpeople | U | −2.6799 | −2.7279 | 5.0 | | 1.81 | 0.071 |
| | M | −2.6798 | −2.7092 | 3.1 | 38.8 | 1.13 | 0.259 |
| lnfis | U | −2.0714 | −2.1343 | 12.7 | | 4.57 | 0.000 |
| | M | −2.0712 | −2.0502 | −4.2 | 66.7 | −1.52 | 0.127 |
| lnsci | U | 11.538 | 11.195 | 27.3 | | 9.80 | 0.000 |
| | M | 11.539 | 11.577 | −3.0 | 88.9 | −1.11 | 0.266 |
| lnurba | U | −1.8564 | −1.7648 | −13.9 | | −5.00 | 0.000 |
| | M | −1.8568 | −1.8376 | −2.9 | 79.0 | −1.06 | 0.289 |

Notes: In the second column, U refers to the sample before employing the PSM, while M denotes the matched sample after applying the PSM.

**Table A2.** Balance Test of Calliper Matching.

| Variables | Unmatched Matched | Mean Treated | Control | Bias (%) | Reduce Bias (%) | t-Test T | P > T |
|---|---|---|---|---|---|---|---|
| lnpeople | U | −2.6799 | −2.7279 | 5.0 | | 1.81 | 0.071 |
| | M | −2.6800 | −2.7130 | 3.5 | 31.1 | 1.27 | 0.204 |
| lnfis | U | −2.0714 | −2.1343 | 12.7 | | 4.57 | 0.000 |
| | M | −2.0710 | −2.0606 | −2.1 | 83.4 | −0.76 | 0.445 |
| lnsci | U | 11.538 | 11.195 | 27.3 | | 9.80 | 0.000 |
| | M | 11.538 | 11.554 | −1.3 | 95.4 | −0.46 | 0.643 |
| lnurba | U | −1.8564 | −1.7648 | −13.9 | | −5.00 | 0.000 |
| | M | −1.8565 | −1.8474 | −1.4 | 90.2 | −0.50 | 0.618 |

Notes: In the second column, U refers to the sample before employing the PSM, while M denotes the matched sample after applying the PSM.

**Table A3.** Balance Test of Kernel Matching.

| Variables | Unmatched Matched | Mean Treated | Control | Bias (%) | Reduce Bias (%) | t-Test T | P > T |
|---|---|---|---|---|---|---|---|
| lnpeople | U | −2.6799 | −2.7279 | 5.0 | | 1.81 | 0.071 |
| | M | −2.6798 | −2.7059 | 2.7 | 45.6 | 1.01 | 0.315 |
| lnfis | U | −2.0714 | −2.1343 | 12.7 | | 4.57 | 0.000 |
| | M | −2.0712 | −2.0654 | −1.2 | 90.8 | −0.42 | 0.672 |
| lnsci | U | 11.538 | 11.195 | 27.3 | | 9.80 | 0.000 |
| | M | 11.539 | 11.531 | 0.6 | 97.8 | 0.22 | 0.826 |
| lnurba | U | −1.8564 | −1.7648 | −13.9 | | −5.00 | 0.000 |
| | M | −1.8568 | −1.8391 | −2.7 | 80.7 | −0.98 | 0.328 |

Notes: In the second column, U refers to the sample before employing the PSM, while M denotes the matched sample after applying the PSM.

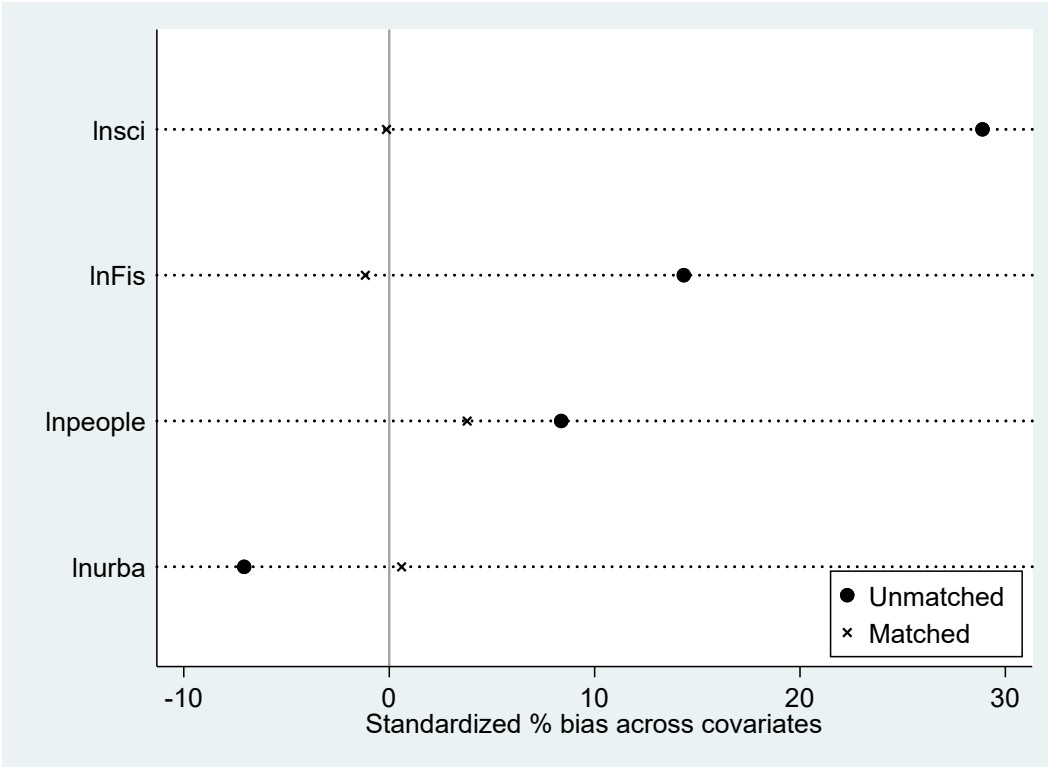

**Figure A1.** Comparison between samples before and after applying nearest neighbor matching.

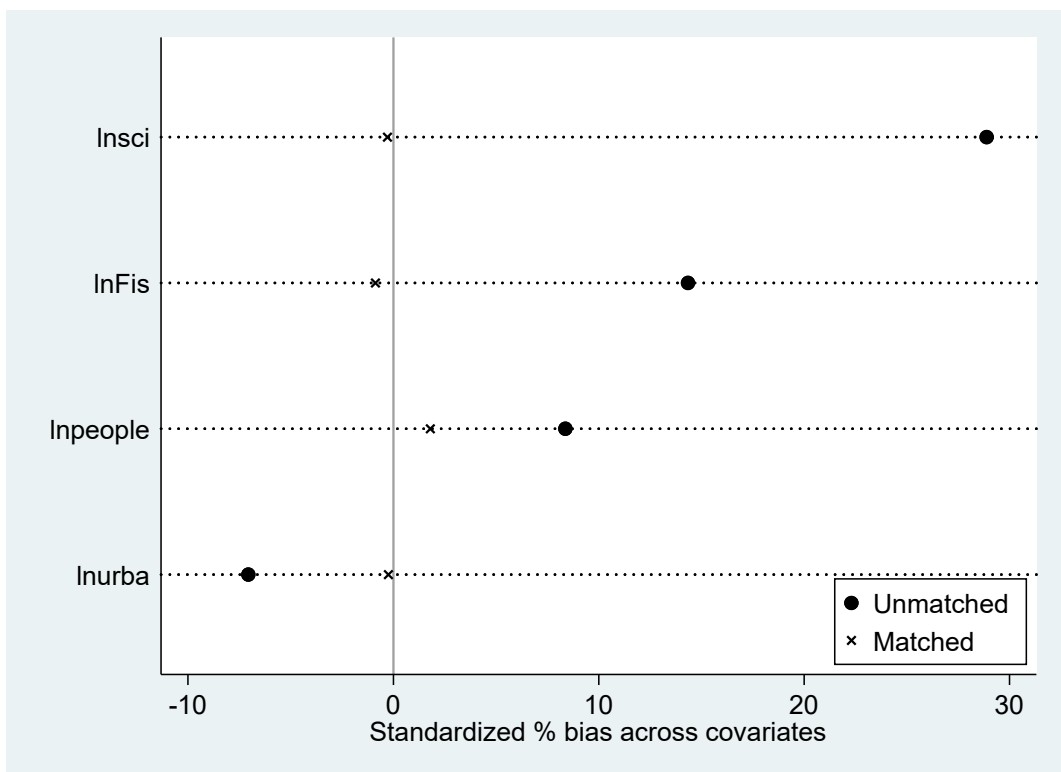

**Figure A2.** Comparison between samples before and after applying calliper matching.

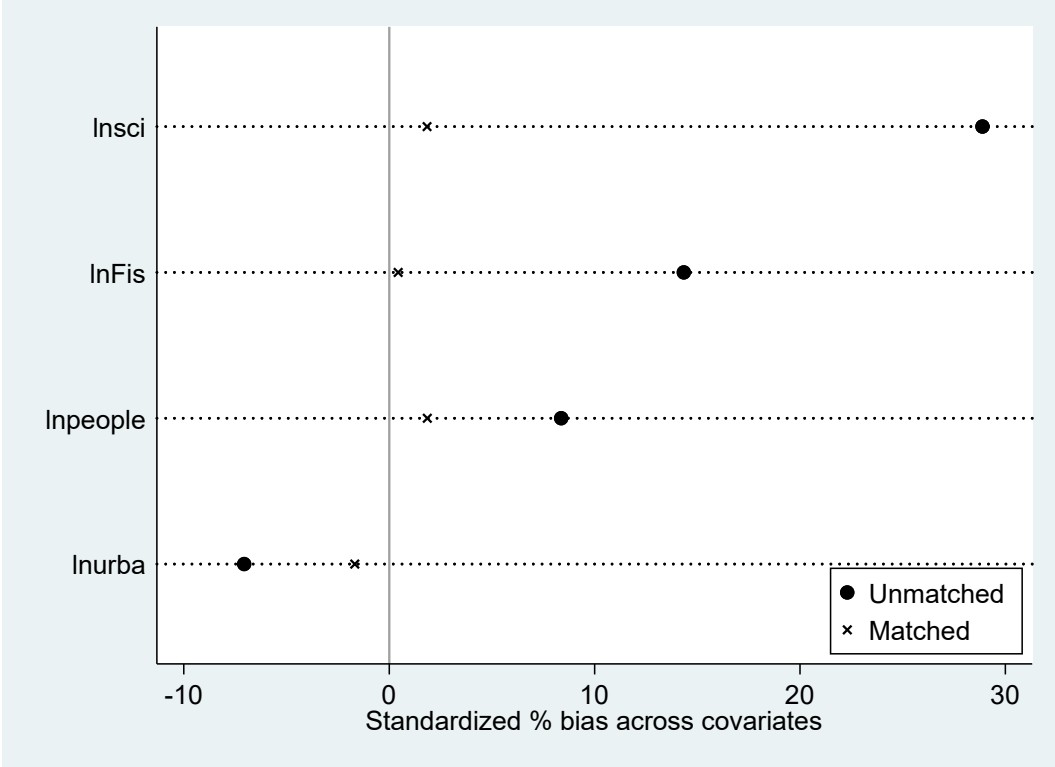

**Figure A3.** Comparison between samples before and after applying kernel matching.

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
