# Peer review of "The Impact of Polycentric Structure on CO2 Emissions: Evidence from China"

_applsci, doi:10.3390/app13105928_

Round 1

Reviewer 1 Report

Overall the paper has done solid research and analysis on CO2 emission polycentric structure in China. I have some suggestions:

1. In 2.2 data summary section, please explain all variables shown in table 2. 

2. 2.3 methodology section is not sufficient and does not fully cover all methodologies used in this paper. For example, robustness check method should be included in 2.3 instead of in 3.2.

3. Please change typos, for example, page 2, line 61, "Chinese".

Author Response

Response: We totally agree with your advice. We added descriptions of variables mentioned in Table 2 as follows (line 121):

Table 1. Definitions and operations of variables.

Variable

Definition

Dependent variable

Carbon

The logarithm of regional CO2 emissions.

lndependent variables

Q

The Pareto exponent.

Mediating variable

Gti

The logarithm of the annual number of green patent applications plus one.

Energy

The logarithm of electricity consumption.

Control variables

lngy

The share of the secondary sector in GDP.

lnFDI

The logarithm of the ratio of the region's actual use of foreign capital to GDP in that year.

lnpeople

The logarithm of the ratio of the total population to the local land area at the end of the year.

lnfis

The logarithm of the ratio of regional general budget fiscal expenditure to GDP.

lnsci

The logarithm of the ratio of regional science and technology expenditure to GDP.

lnurba

The logarithm of the ratio of the urban employed population to the year-end household population.

Other variables

Hlhl

The ratio of the river density to the exchange rate.

Unemploy

The ratio of unemployment to the population.

LCC

If it is the pilot city of Low-carbon city policy, this variable = 1, 0 otherwise.

ETS

If it is the pilot city of the emissions trading scheme, this variable = 1, 0 otherwise.

Rep_carbon

The CO2 emissions proposed by of Wu et al. (2016) [17]

PCE

CO2 emissions per capita.

Lngreencov

The logarithm of canopy coverage.

Lnlpg

The logarithm of household consumption of liquefied petroleum gas.

Lnws

The logarithm of average wind speed.

Lnhum

The logarithm of average relative humidity.

Lntem

The logarithm of average temperature.

  1. 2.3 methodology section is not sufficient and does not fully cover all methodologies used in this paper. For example, robustness check method should be included in 2.3 instead of in 3.2.

Response: Thanks for your suggestions. We reorganized the section 2.3 as follows (line 136-169):

“To mitigate concerns about endogeneity and figure out the causality between polycentric spatial structures and CO2 emissions, we constructed two IVs that were correlated with polycentric spatial structures but met the exogeneity condition of CO2. Specifically, we followed Chen et al. (2021) and used the river density/exchange rate (i.e., Hlhl) as the first instrument for polycentricity [10]. Proximity to water sources could affect the spatial structure of cities [18]. In the meantime, it is not related to CO2 emissions. Thus, river density is a proper IV for polycentricity.

We then proposed an innovative instrument, namely, the proportion of unemployment (i.e., Unemploy, unemployment as a percentage of the population). It has been proved that there are interdependencies between labour market behaviour and the spatial structure of cities [19]. A large proportion of unemployed people indicates that the city has developed excessively or poorly, which leads to the lack of impetus for further expansion of the city, which has a negative impact on polycentricity. Meanwhile, the proportion of unemployment cannot affect CO2.

Finally, the ratio of river density to exchange rate and the proportion of unemployment are IVs for polycentric spatial structure. The first stage of two-stage least square (2SLS) were as follows:

(3)

where  and  represent the ratio of river density to exchange rate and the proportion of unemployment, respectively.

The second stage of 2SLS were as follows:

(4)

where  is the first stage predicted value for the polycentricity (i.e., ).

Previous studies have verified the mediating role of transportation, household carbon emissions, and industrial emissions in the relationship between polycentric structures and CO2 emissions [6,11]. We proposed two innovative pathways to examine how polycentric structures affected CO2 emissions. We hypothesized that green technology and energy consumption might serve as potential mechanisms through which polycentric structures influenced CO2 emissions. To test these hypotheses, we measured green technology as the logarithm of the annual number of green patent applications plus one assessing its mediating effect on the relationship. For energy consumption, we use the logarithm of electricity consumption to evaluate its potential mediating role in the relationship between polycentricity and CO2 emissions. With these variables in place, we constructed the following model to analyze the impact of polycentric structures on CO2 emissions through these proposed pathways:

(5)

where  represents mediators (i.e., green technology, and energy consumption).”

  1. Please change typos, for example, page 2, line 61, "Chinese".

Response: Thanks for pointing out this. We correct the typos here. To guarantee the language quality, we also polished the language in this study (the revisions are highlighted in yellow).

Reviewer 2 Report

The manuscript entitled "The impact of polycentric structure on CO2 emissions: Evidence from China" provides the importance of using polycentric structure towards the reduction of CO2 emissions. As far as the novelty is concerned, it is very much novel idea and the idea was executed very well. The Introduction section is very much planned and creating a nice set of information that can even understood by a lay man. The Experimental section, results and their analysis, discussion are upto the mark. My point of disappointment is the lack of scientific information, as the authors are linking the morphology to the CO2 efficiency and so I wanted to see some materials structures. However, all the information provided is statistical analysis and not even a single results related to materials science, physics or chemistry. As I looked at the journal and it is "Applied Sciences" and I wanted to see some scientific data. Some scientific data and its related statistical analysis are fine to me. Overall, I have no doubt to say that the idea applied in this manuscript is very much novel and executed very well. My only concern is the aim/scope of the journal linking to the information presented in the manuscript. Therefore, I promote this manuscript to publish in the present form as long as the editors don't have any issue to publish in this particular journal, where many other statistical journals are available from the same mdpi publisher.  

Author Response

Response: Thanks for your suggestions. We revised the introduction and conclusion to better fit the aims and scope of this special issues. The revisions are as follows:

Revisions in Introduction section (line 28-51):

“  The 2030 Agenda for Sustainable Development offers a visionary roadmap for a sustainable future. There is global consensus on coordinating spatial expansion, economic growth, and environment to achieve the sustainable development [1,2]. Sustainable Development Goals 11 (SDG 11) aims to create inclusive, safe, resilient, and sustainable cities and communities for all. However, the increasing threat of climate change looms over our living environments. The most recent progress report on SDG 13, which focuses on climate action, highlights the urgent need to reduce CO2 emissions, as they reached a historic high in 2021 [3]. In countries like China, rapid urbanization and the concentration of resources in major cities exacerbate CO2 emissions. Consequently, finding a balance between resource integration and sustainable development has become a critical challenge in urban planning.

Research has shown that adopting a polycentric structure can be beneficial in terms of resource integration and optimizing efficiency [4]. Thus, clarifying the impact of polycentric structure on CO2 emissions can help to direct the practice of sustainable development. However, the relationship between polycentric structure and CO2 emissions has not reached a consensus. Some studies prove that the polycentric structure takes advantage of the integration of resources, and optimize the efficiency of factors [5]. Thus, it can alleviate CO2 emissions [4,6]. It's important to note that some studies have raised concerns about polycentricity, suggesting that it may lead to higher household emissions, increased industrial manufacturing [7], and greater household heating demands [8], all of which could contribute to higher CO2 emissions. While the other studies demonstrate a non-linear relationship between polycentric structure and CO2 emissions [9,10]. This highlights the need for further exploration and discussion to better understand the intricacies of this relationship.”

Revisions in Conclusion section (line 361-392):

“Fuelled by the urgency of achieving Sustainable Development Goals (SDGs), ad-dressing the rise in CO2 emissions has become a pressing concern. This study delves into the role of sustainable urban development in mitigating CO2 emissions by investigating the impact of polycentric structures on CO2 emissions in 278 Chinese cities from 1999 to 2017. We employed a two-way fixed effect model to estimate our benchmark results and constructed instrument variables (IVs) to address endogeneity concerns. Our findings reveal that polycentric structures contribute to CO2 reduction by promoting green technology and lowering energy consumption. Notably, the positive impact of poly-centric structures on CO2 emissions is only significant in cities that are not classified as municipalities or resource-based cities.

Our study marginally contributes to the theory. First, we figure out a robust relationship between polycentric structures and CO2 emissions, accounting for potential biases from, policy interferences, selection bias, and core variable measurements. Second, we propose two innovative pathways that mediate the impact of spatial structures on CO2 emissions. Third, we consider the heterogeneous effects of administrative hierarchy and resource dependence in this context.

From our findings, we derive practical implications for achieving SDGs. First, polycentric structures offer a viable approach for developing countries to regulate CO2 emissions. When it comes to China specifically, some cities in China should gradually build polycentric spatial structures. Second, government departments should implement strategies related to green technology and energy consumption to regulate CO2 emissions and achieve SDGs. Third, urban planning and CO2 mitigation strategies should be tailored to individual cities, with a focus on establishing more environmental regulations in resource-based cities and municipalities.

In closing, we acknowledge some limitations and avenues for future research. First, our study uses county-level data to calculate the morphological polycentric index at the city level, overlooking the functional dimensions of polycentric structures. Future studies should explore these functional dimensions further. Second, our analysis focuses on the Pareto index in relation to city size balance, suggesting that future research could examine the impact of spatial balance on CO2 emissions. Lastly, while our study area is China, future studies could investigate the heterogeneous effects of diverse countries and delve into other aspects of SDGs to gain a more comprehensive under-standing of the topic.”

Reviewer 3 Report

The article deals with important issues of CO2 emissions into the atmosphere. The purpose of the research is clearly defined. The methodology, although quite commonly used, does not raise any objections. The results are very interesting but easy to predict.

To sum up, in my opinion, the article, although not innovative, can be published.

Author Response

Response: Thanks for your suggestions. We revised the introduction, method, discussion, and conclusion sections to make this study more relatable and engaging. All the revisions are highlighted in yellow.

Reviewer 4 Report

Very interesting and a crucial subject area. 

the introduction is well written. The objectives of the study is stated as to "evaluate the impact of morphological poly centric structures on CO2 emmisions".  

You have selected a set of control variables focused on socio economical aspects. when you consider urban contexts, there are other physical parameters that could influence the measures carbon emmissions.  for example, if there is a large canopy cover, that could has a different implication, to a setting with no canopy cover even if all the other socio economical aspects remain the same. climatic condition will also have a similar impact, physically and driving energy consumption behaviour. I would suggest to include this into the study. Otherwise the conclusions are not valid. 

Also suggest to develop a conceptual framework, that will guide the research and repenting the variables in a graphical format. This can be a useful contribution for other scholars to replicate the study. 

Author Response

Response: We totally agree with your advice. Physical parameters could also impact carbon emissions. Thus, we added a section in Robustness checks to mitigate the influences of these confounders. We added the section 3.2.6 as follows (line 266-282):

“3.2.6. Exclude other confounders

Despite accounting for various confounding factors that could influence the relationship between polycentric structures and carbon emissions in our baseline results, there are still some potential confounders that we have not considered. Factors such as canopy coverage [25], energy consumption [26], wind speed [27], humidity [28] and temperature [29] may affect carbon emissions. To minimize the impact of these confounders, we included them as controls in our baseline results. Columns (1)-(5) in Table 6 displayed the effect of polycentric spatial structures on CO2 emissions after controlling for canopy coverage, energy consumption, wind speed, humidity, and temperature, respectively. The results indicated that polycentric structures reduce carbon emissions even after accounting for the confounding influences of these additional factors. Column (6) presented the results after controlling for all potential confounders, and the findings remained robust.

Table 6. Robustness check: The impact of polycentric spatial structures on CO2 emissions

(1)

(2)

(3)

(4)

(5)

(6)

Variables

Carbon

Carbon

Carbon

Carbon

Carbon

Carbon

Q

-0.052***

-0.051**

-0.050***

-0.050***

-0.050***

-0.051***

(-2.77)

(-2.55)

(-2.60)

(-2.60)

(-2.59)

(-2.67)

lngreencov

0.030

0.031

(1.43)

(1.38)

lnlpg

0.003

0.002

(0.004)

(0.41)

lnws

-0.005

-0.014

(0.047)

(-0.28)

lnhum

-0.162*

-0.162*

(-1.91)

(-1.81)

lntem

0.022

-0.007

(0.85)

(-0.30)

Control

YES

YES

YES

YES

YES

YES

City FE

YES

YES

YES

YES

YES

YES

Year FE

YES

YES

YES

YES

YES

YES

Observations

4,473

4,400

4,513

4,513

4,510

4,364

R-squared

0.948

0.947

0.947

0.947

0.947

0.948

1 t-values are in parentheses; *, **, and *** represent the 10%, 5%, and 1% significance levels, respectively; the coefficients of the control variables (i.e., the secondary industry as a percentage of GDP, foreign direct investment, population density, the percentage of budget fiscal expenditure and technology expenditure on GDP, and the ratio of the employed population), year and time dummies are not reported to save space.”

Furthermore, we added a conceptual framework in the Introduction section as follows:

“The conceptual framework is represented in Fig.1”.

Round 2

Reviewer 4 Report

The conceptual framework improved the clarity of the paper. Thank you for taking the feedback constructively and revising the methodology to include additional confounders.